# Trends and Characteristics of Human Casualties in Wildlife–Vehicle Accidents in Lithuania, 2002–2022

**DOI:** 10.3390/ani14101452

**Published:** 2024-05-13

**Authors:** Linas Balčiauskas, Andrius Kučas, Laima Balčiauskienė

**Affiliations:** 1Nature Research Centre, Akademijos 2, 08412 Vilnius, Lithuania; laima.balciauskiene@gamtc.lt; 2Joint Research Centre, European Commission, Via Fermi 2749, 21027 Ispra, Italy; andrius.kucas@ec.europa.eu

**Keywords:** human casualties, transports accidents involving wildlife, moose, temporal trends, wildlife fencing

## Abstract

**Simple Summary:**

Analysis of 474 human casualties in wildlife–vehicle accidents (WVAs) in Lithuania between 2002 and 2022 revealed that their numbers have been escalating since 2018. Moose were the primary contributors to fatalities and injuries despite their much lower abundance in the country compared to that of roe deer. Temporal patterns showed that most casualties occurred during dusk or dawn in May and September, on weekends, and between 20:00 and 22:00. Spatially, main roads with high traffic density had the highest casualties per unit length. Most casualties occurred after vehicles directly hit an animal with cars and motorcycles being the most vulnerable. The effectiveness of WVA prevention measures was inconclusive with a small percentage of casualties occurring in areas with warning signs or fenced road segments. These findings highlight the need for a critical evaluation of current prevention strategies to reduce human casualties in WVAs.

**Abstract:**

We analyzed 474 human casualties in wildlife–vehicle accidents (WVAs) that occurred between 2002 and 2022 in Lithuania, which is a small northern European country. The study revealed the escalating trend of WVAs, since 2018 surpassing other transport accidents, although the number of casualties per WVA was ca. 100 times lower compared to other transport accidents. Moose was the primary contributor, responsible for 66.7% of fatalities and 47.2% of injuries, despite much lower species abundance compared to roe deer, which is the main species involved in WVAs without human casualties. Temporal patterns highlighted seasonal, daily, and hourly variations, with the majority of casualties occurring during dusk or dawn in May and September, on weekends, and between 20:00 and 22:00. Spatially, main roads with high traffic density exhibited the highest casualties per unit length. Most casualties occurred after hitting an animal directly with cars and motorcycles being most vulnerable vehicles. The effectiveness of WVA prevention measures was inconclusive: 9.5% of fatalities and 1.4% of injuries were registered in the area of the warning sign, and 10.4% of all casualties occurred on fenced road segments. These findings suggest the need for a critical evaluation of the current prevention strategies in reducing human casualties associated with WVAs.

## 1. Introduction

Road traffic accidents cause about 1.35 million human deaths annually [1]; another 50 million people sustain injuries [2]. Despite initiatives taken to reduce human mortality on roads and the implementation of road safety measures [3], the number of fatalities recently increased not only in low-income and middle-income countries but also in the developed ones [4].

Three significant initiatives to reduce human casualties during transport accidents were adopted in the last few years. One is the UN resolution “Improving global road safety”, adopted by General Assembly in 2020, with a plan to reduce road traffic deaths and injuries by 2030 [5]. The World Health Organization set a Global Plan for the Decade of Action for Road Safety in 2021 [6]. Both of these, however, do not include a reduction in the number of wildlife–vehicle accidents (WVA) on the roads [2].

According to the number of persons killed in road traffic accidents per million inhabitants in 2016, Lithuania, with 66.0, is above the EU average of 50.5 [3]. In 2022, 20,640 people died in the EU as a result of road accidents with an increase of 4% compared to 2021. As compared to 2019, two countries showing the biggest decrease were Poland and Lithuania, which were both down 35% [7]. Ultimately, Lithuania emerges as the leader in the European Union for reducing the number of people killed in road accidents in the last decade, achieving a remarkable −60% decrease between 2012 and 2022 [8].

The ambitious goal of “no kills or injuries within the road transport system” is implemented in the Vision Zero [9,10]. These priorities correspond to similar EU standards [11]. In the US alone, the annual number of human deaths due to wildlife–vehicle accidents (WVAs) exceed 440, while the number of injuries was over 59,000 [12]. As for WVAs in Lithuania, Vision Zero foresees a drop in the number of such accidents on state roads to five per year, which would be a decrease of 83% [13]. Various countries approach the target of WVA mitigating by means of directional fencing [14] and wildlife passages [15] as well as planning species connectivity networks at the landscape level [16].

WVA are not the main source of human road casualties, but they also affect animals [17]. In Brazil, the estimated annual animal roadkill is about 475 million animals [18] with the number of human casualties unreported, although 18.5% of police-registered WVAs resulted in fatalities [19]. A summary for Europe shows about 223 million WVAs annually, causing about 300 human deaths and 30,000 injuries, with damages estimated at one billion euros [20]. Information of human casualties from the Baltic countries (Lithuania, Latvia and Estonia) is not available publicly.

In the earlier decades, WVA investigations focused on the main roads [21,22,23], and recently, problems have arisen even in the urban areas [24,25] and require attention. As WVAs occur also on the low traffic intensity roads [26,27], wildlife fencing is not a universal solution [28].

Therefore, the driver factor has received a lot of attention recently, although most of these studies are not related to WVAs [29,30]. Driving style, such as speed or acceleration, and approach to security measures were shown as the factors of the risk of crash [31]. Driver behaviors complement other risk factors, which are related to the environmental and road characteristics, WVA hotspots and temporal trends in animal behavior, and transport intensity [32].

It is possible to reduce or minimize human-related accident factors, but it is not cost-effective, if not impossible, to eliminate wildlife factors on a large scale. While the effectiveness of the wildlife fencing system has been demonstrated on major roads [33], its implementation on all roads throughout the country is impractical [28].

Therefore, instead of wildlife exclusion from the road, early warning of the driver and understanding driver attitudes and habits might be a good perspective [30]. Automated warning systems, allowing braking to commence even before a deer is visible to the driver, are currently under development and testing [1]. However, integrating artificial intelligence into driver assistance or automated driving systems presents challenges [34]. Ethical considerations are necessary when determining a course of action for unavoidable collisions with wildlife [35].

In Lithuania, three programs for road safety were implemented in the past, none of which were related to WVA problems [13]. Road Safety Audits in 2011–2017 revealed that 16.0–39.4% of problems found were related to the road signs and vertical marking, which are the main problem on the main, national and regional roads [4].

There are four priorities in the new program “Vision–zero in transport in 2018–2050” in Lithuania: (1) Safer Behavior of Road Users, (2) Safer Roads, (3) Safer Vehicles, and (4) More Efficient Rescue Assistance After a Road Crash. The second one includes the development of road infrastructure ensuring the safer movement of animals when crossing the road network. It aims to “…increase the length of fences against wildlife, the number of wildlife crossings and other roadside protection measures… with the aim to reduce the number of encounters with large and small species of wildlife” [13]

From Lithuania, the only analysis of human casualties in transport accidents involving animals in 2014–2018 is given in [36], but the authors could not present models due to limited sample sizes. Naidič and Bražiūnas [37], when analyzing the dynamics and causes of road fatalities in Lithuania, did not provide a WVA category, although WVA may be hidden under the categories “unexpected obstacles” and “other”.

The other traffic accident-related scientific analyses from the country do not include a wildlife component. For example, comparing transport accident patterns in cities of Lithuania and Sweden [38], the authors do not mention the WVA problem, though it is really threatening [24]. Other papers analyze driver-related aspects [39] and/or temporal and meteorological factors but not in the WVA context [40].

Therefore, we argue that even in a smaller country, there is a need to tackle the important problem of wildlife–vehicle accidents (WVAs), which result in fatalities or injuries to drivers and/or passengers. The aim of this paper is to analyze the spatiotemporal characteristics of human casualties due to wildlife–vehicle accidents in Lithuania from 2002 to 2022. We cover several aspects of WVA-related human casualties, such as (1) their proportion in the total toll of all transport accidents, (2) influence of the wildlife fencing on the occurrence of human casualties, (3) wildlife species involved, and (4) driver-related aspects.

## 2. Materials and Methods

### 2.1. Study Area

The study area covered the territory of Lithuania (Figure 1), a small country in northern Europe with a surface area of 65,300 km^2^, a population of 2,857,279 in 2023, and a human population density of 45.3 inhabitants per km^2^ [41]. Agricultural lands cover 52.6% of the country, forests 33.2%, built-up territories 3.6% and roads 1.6% [42,43].

The road network in the country consists of 1750.71 km of main roads with an annual average daily traffic (AADT) rate of about 10,000 vehicles/day, 4927.68 km of national roads with an AADT of up to 3000 vehicles/day, and 14,559.24 km of regional roads with an AADT of up to 500 vehicles/day [44,45].

Wildlife fencing in the country was mostly installed by 2017, as it was used as a main WVA mitigation measure. Currently, 3.8% of all roads in the country are protected by wildlife fencing. Most of the fences (743.8 km out of 803.5 km, or 680 out of 1088 segments) are constructed on the main roads [28]. Other WVA mitigation effects, such as underground wildlife passages, one-way gates, and wildlife guards, are used sparingly [27].

### 2.2. Data Collection

Data on human mortality and injuries were aggregated from the Lithuanian Road Police Service webpage [46]. During 1998–2022, there were 112,641 transport accidents, causing 11,154 human deaths and 135,638 human injuries.

Recording the number of the wildlife–vehicle accidents (WVAs) was started by the Lithuanian Police Traffic Surveillance Service in 2002. During 2002–2022, the number of WVAs was 45,653. In these accidents, 21 WVAs resulted 22 human deaths, while the number of injuries in 360 WVA cases was 452. However, these data are depersonalized; therefore, the number of lethal outcomes of WVAs was reported in the on-site reports, but no check of the fate of injured persons in hospitals was made.

### 2.3. Data Analyses

Due to the limited annual samples of human casualties, especially fatalities, a multivariate analysis was deemed impractical. Instead, we opted for a comparative approach, focusing on proportions. We compared the proportions of animal species involved in WVAs with human casualties, considering the distribution of casualties by month, day of the week, phase of day, hour, and light levels, categorized by road type. This analysis also considered the presence of wildlife fencing, mode of transport, and various driving-related aspects. 

We utilized the 95% CI calculated in OpenEpi [47] using the Wilson method, known for its accuracy and suitability, especially for small samples or extreme proportions [48]. Differences in proportions were assessed through the G test [49].

Using Pearson’s correlation coefficient and choosing the best regression based on Akaike’s IC [50], we tested whether the number of injured persons per WVA correlates with the body mass of the animal killed on the road. Data on the average body mass of animals were collected from [51,52,53,54]. Correlations of human casualties per km of the main road with AADT and road length were also tested. 

The influence of the wildlife fencing was analyzed using following scenario: fence was not present at the time of the WVA, fence present and the WVA occurred inside the fenced segment of the road, fence present and the WVA occurred within 50 m from the end, and the WVA happened outside the fenced area.

Three periods of varying light intensity were distinguished depending on the time of day: night, dusk/dawn and day. The twilight period was defined as the hour before and the hour after sunset and sunrise. The times of sunrise and sunset, depending on the date, were taken from [55]. 

Average numbers of casualties per day of week were compared using Student’s *t*-test. All calculations were made using PAST ver. 4.13 [50].

Human casualties were geo-referenced using either (i) the address of the collision or (ii) the location of a collision indicated by road number and distance from the start of the road, and/or (iii) geographical coordinates of the collision. The ESRI Inc. ArcGIS^®^ Desktop ver. 10.8.1 software was applied for spatial data interoperability, collision geo-localization, and mapping.

## 3. Results

### 3.1. Human Casualties in Transport and Wildlife-Related Accidents

The number of transport accidents in Lithuania from 1998 to 2007 remained stable with an annual average of about 6300. Subsequently, the number of transport accidents exhibited a decreasing trend, dropping from 4800 in 2008 to approximately 2800 per year in the period of 2020–2022 (Figure 2).

The number of human casualties is highly correlated with the number of transport accidents with the Pearson correlation coefficient being r = 0.986 for lethal outcomes and r = 0.999 for cases of injuries. Therefore, trends in human casualties mirror those of transport accidents, remaining stable in the period 1998–2007 and decreasing from 2008 onwards.

The number of fatalities per accident also decreased: in 1998–2007, it was 0.117 dead and 1.225 injured; after 2008, it dropped to 0.074 and 1.174, respectively.

On the contrary, the number of registered WVAs in Lithuania from 2002 to 2022 demonstrates a linear increase with the exception of 2017 and 2020. The drop in 2020 can be attributed to restricted human mobility during the COVID-19 pandemic [56]. In 2002, the number of registered WVAs was a mere 376, constituting 6.2% compared to the number of transport accidents. In 2017, there were 2423 registered WVAs, accounting for 79.4%, and from 2018 onwards, the number of WVAs surpassed the number of transport accidents (Figure 3).

However, this increased number of WVAs does not correlate with the number of human fatalities in such accidents (r = 0.041, NS). Instead, it strongly correlates with the number of injuries (r = 0.800, *p* < 0.001). The relative numbers of fatalities remained quite steady, averaging 0.7 per 1000 WVA, with a maximum of 4 in 2007 (Figure 3). The average number of injured persons per 1000 WVAs was 13, and it has been decreasing in the last five years, which was possibly influenced by the larger number of WVAs.

### 3.2. Involved Wildlife Species

Four wild and domestic mammal species were involved in WVAs resulting in human fatalities, with moose being the cause in 66.7% of such cases (Table 1). This was followed by roe deer and horses, each accounting for 9.5% of incidents. In cases of WVA ending with human injuries, a total of 12 species were involved. Moose once again had the highest proportion at 47.2%, which was followed by roe deer (13.6%), cattle (6.4%), horse (6.1%), and red deer (4.7%). So, in both human fatalities and injuries, moose dominated among wildlife species (G = 25.1 and G = 272.1, respectively, *p* < 0.001).

The number of injured persons per WVA was the highest after accidents involving European bison, which was followed by those with cattle, horses, moose, and red deer (Table 1). The number of injured individuals per one accident was linearly related (AIC = 5.388) to the body mass of the road-killed animal with a correlation of r = 0.96 (*p* < 0.001). 

### 3.3. Temporal Distribution of WVA-Related Human Casualties

Human casualties were not equally distributed by month (G = 116.4, *p* < 0.0001). In May and September, the proportion of fatalities was significantly higher than the monthly average (G = 44.3, *p* < 0.0001), while months from December to March were characterized by a smaller number of casualties (Table 2).

The distribution of human casualties during the week (Table 3) was also not even (G = 15.9, *p* < 0.05). The highest proportion of casualties was registered during weekends, from Friday to Sunday, exceeding the average (G = 10.3, *p* < 0.05). The average number of fatalities per workday did not differ significantly from that on weekdays (2.5 vs. 3.7, t = 0.65, *p* = 0.54), but the average number of injuries per weekend day was significantly higher (43.8 vs. 60.3, t = 4.24, *p* < 0.01).

Most of the fatalities, constituting 42.9% of the total cases, occurred within a narrow time interval from 20:00 to 22:00, corresponding to dusk/dawn or night conditions depending on the time of year (Figure 4). The distribution of injuries was associated with two distinct periods: evening (17:00–00:00, 57.0%) and morning (06:00–07:00, 9.8%).

Regarding light conditions, the majority of injury cases occurred during dusk and dawn, accounting for 46.6% (CI = 41.5–51.8%) of all cases, significantly surpassing those that occurred at night (30.1%, CI = 25.5–35.1%, G = 20.1, *p* < 0.001). In the daytime, 23.3% (CI = 19.2–28.0%) of all WVAs resulting in human injuries were recorded.

### 3.4. Influence of the Road Type and Spatial Distribution of Human Casualties

Most of the human casualties due to WVAs occurred on the main roads of the country (Table 4). However, differences in the number of fatalities by road type were not significant (G = 4.0, *p* = 0.15), while the number of injuries differed (G = 136.5, *p* < 0.001), being lowest on regional roads. During the period 2002–2022, the number of human casualties per 1000 km of main roads was four times higher than that on national roads and 36 times higher than that on regional roads.

Human fatalities due to WVAs were concentrated on two main roads of the country, A1 and A2 (Figure 5), which connect the four largest cities and are characterized by the highest AADT. Out of the 11 lethal WVA incidents on the main roads, 10 (90.9%) were due to collisions with moose. On regional roads, WVAs with moose resulted in three out of six (50.0%) human fatalities.

Out of 154 cases of human injuries on the main roads, 101 (65.6%) resulted from WVAs with moose, which was followed by 11 (7.1%) from WVAs with roe deer (Figure 6). In 14 (9.1%) WVA cases, the animal species was not identified, and cattle and red deer each had 8 (5.2%) WVA incidents.

Out of 112 cases of human injuries on the national roads, the majority resulted from WVAs with moose—42 (37.5%), followed by 21 (18.8%) WVAs with roe deer, 12 (10.7%) with horses, 9 (8.0%) with cattle, and 8 (7.1%) with red deer. In 11 WVA cases (9.8%), the animal species was not identified.

On the regional roads, out of 32 cases of human injuries in WVAs, the majority (8 cases, 25.0%) involved moose, which was followed by those with roe deer (6 cases, 18.8%) and cattle (5 cases, 15.6%).

On urban roads and streets, the highest number of human injuries was registered after WVAs with domestic dogs (18 out of 58, 31.0%), which was followed by moose (10 cases, 17.2%) and roe deer (8 cases, 15.5%). In 10 WVA cases, the animal species was not identified.

Of the total number of WVAs resulting in human injuries on main roads, three of them, A1, A2, and A11, represented 57.8%. Six main roads, A4, A6, A10, A12, A13, and A16, accounted for 30.5%, while the remaining eight main roads constituted only 11.7% of cases (Table 5). The number of injury cases on the main roads correlates with the road length (r = 0.76, *p* < 0.001) and AADT of cars (r = 0.63, *p* < 0.01) but not the AADT of trucks (r = 0.12, NS).

During 2002–2022, the highest number of human injuries per kilometer on the main road occurred on A1, A2, A11, and A13. This number correlates with the AADT of cars (r = 0.50, *p* < 0.05) but not with the road length (r = −0.43, *p* < 0.10) or AADT of trucks (r = −0.09, NS).

### 3.5. Influence of Wildlife Fencing and Warning Signs

The locations of human casualties in WVAs with relation to the presence of wildlife fencing are presented in Figure 7. In 72 cases, the WVA occurred prior to the fence construction—these cases are still shown; therefore, it may appear that more casualties are located in the fenced segments of the roads.

However, only 30 cases (10.4%, 95% CI = 7.4–14.5%) of all human casualties after a WVA actually occurred inside the fenced area, while three cases (1.0%, 95% CI = 0.3–4.0%) were within a distance of less than or equal to 50 m from the end of the wildlife fence. The remaining 255 cases (88.5%, CI = 84.3–91.7%) occurred more than 100 m from the end of the fence. Confidence intervals do not overlap, and the G-test shows significant differences between groups (G = 661.1, *p* < 0.001) with prevailing locations of human casualties outside the fenced areas (G = 547.3, *p* < 0.001).

As for fatalities, in 8 out of 21 cases, the fence was constructed later than when the WVA happened. When the fence was present, in four cases (30.8%, CI = 12.7–57.6%), human fatality took place inside, while in nine cases (69.2%, CI = 42.4–87.3%), it took place outside the fenced segment. These proportions did not differ significantly (G = 2.5, *p* = 0.12). All four human fatalities inside the wildlife fences were due to WVAs with moose.

As for human injuries, in 64 out of 342 cases, the wildlife fence was constructed later than when the WVA occurred. The distribution of the remaining injury cases is similar to that of human casualties: 10.1% (CI = 7.1–14.2%) inside the fences, 1.1% (CI = 0.4–3.1%) within 50 m from the end of the fence, and 88.8% (CI = 84.6–92.07%) more than 100 m from the end of the fence. The differences are significant (G = 644.5, *p* < 0.001), and the scenario outside the fence prevails (G = 531.1, *p* < 0.001). Within the fenced segments, 27 out of 30 cases were after WVA with moose, while WVA with roe deer, red deer, and horse were each represented by one case.

The road sign warning about the possible presence of wild animals was present in 2 out of 21 cases (9.5%) with human fatalities and in 5 out of 356 cases (1.4%) with human injuries.

### 3.6. Transport Type and Driving-Related Aspects

The majority of human fatalities (90.5% of cases) were related to car drivers, while fatalities of minibus and truck drivers were each represented by a single case (4.8%). The first category significantly prevailed (G = 43.9, *p* < 0.001).

Most of the injuries were also related to cars (84.8%), which was followed by motorcycles (10.4%). Minor categories included scooters and mopeds (2.2%), bicycles (1.7%), and minibuses (0.8%). All categories differ significantly (G = 1056.6, *p* < 0.001), with the first one significantly exceeding the second, and both significantly exceeding the others (*p* < 0.001).

All fatal WVAs occurred after a direct hit into an animal, which was in some cases followed by a run-off-the-road, pulling into the opposite lane, driving into a tree, etc. (Table 6). A direct hit into an animal with no other sequence of events was the main cause of injuries during WVAs, comprising 78.8% (CI = 74.1–82.6%), significantly exceeding all other scenarios (G = 246.0, *p* < 0.001). In 31 cases, after hitting an animal, the car ran off the road, which was followed by hitting trees or a guardrail, and rollover (Table 6). Thus, in 12.1% of cases, after hitting an animal, the car ran off the road, which was followed by hitting trees or a guardrail, and rollover. In 2.8% of cases, the primary reason was failing to brake while avoiding an accident, and in 1.1%, it was an unsuccessful swerve, which was mainly followed by running off the road. Of the total casualties, 84.8% involved cars, 10.4% involved motorcycles, while vans and trucks were sporadically affected. Drunk driving was involved in only two WVA cases.

## 4. Discussion

We found that the linear increase in the number of WVAs in Lithuania has led to the number of WVAs exceeding the number of other transport accidents since 2018. However, the number of casualties (0.7 fatalities and 13 injuries per 1000 WVA) is much less than in other transport accidents, which are 74 and 1174, respectively. In 97.6–100% of all WVAs, human casualties were not registered with the casualty-free WVA proportion exceeding 99% in 2019–2022.

The proportion of casualty-free WVAs in Lithuania was approximately 10 times larger than reported for Washington State, US, by Al-Bdairi et al. [32]. While WVAs raise road safety and economic concerns [57], a significant proportion of these accidents do not involve complaints and are not registered [58]. Other WVAs are detrimental to both wildlife and humans alike [59]; therefore, they are documented, but reports might lack descriptions of important factors potentially affecting the likelihood of a crash or the severity of consequences [32].

Out of the 12 animal species involved in human casualties in Lithuania, the main cause was moose, responsible for 66.7% of fatalities and 47.2% of injuries, followed by roe deer. These figures do not correspond to the abundance of the species, with 20,676 moose and 172,599 roe deer reported in 2023 as the national population [60], nor with the number of WVAs involving these species, which were recorded as 2708 and 29,256, respectively [61]. We believe that this is due to the different size of these animals (moose are on average 12 times larger than roe deer), which leads to a higher risk of human casualties at collision.

Road accidents involving ungulates pose a significant threat to road safety in Europe [62] and globally [20]. They also interfere with population management [63]. We do not discuss WVA numbers of ungulates in different countries here—it is presented in detail in [61]. Sufficient knowledge of population abundances, distribution, and temporal aspects of accidents is essential for the proper implementation of WVA mitigation measures. These measures must take into account movement patterns and be species-specific [64].

We found three temporal patterns of human casualties in WVAs: (1) seasonal, with the highest numbers in May and September, following the seasonal migration and rut time, and the fewest in December–March; (2) daily, with higher numbers on Friday–Sunday; and (3) hourly, with the highest numbers from 20:00 to 22:00, where most casualties (46.6%) occurred at dusk or dawn—significantly more than at night or during the daytime. The general patterns of ungulate WVAs [65] are similar to those of human casualties. Therefore, knowledge of activity patterns of ungulates is important, as it indicates the probabilities of these animals crossing roads in various countries, which, in general, is crepuscular or nocturnal [66,67,68]. Changes in the long-term trends of ungulate roadkills are related to shifts in their biology. For example, the spring peak of road accidents involving roe deer is now advanced by one month compared to 40 years ago [69], but this has so far been overlooked in the analyses of human casualties in WVAs.

In the spatial aspect, we found that in Lithuania, most WVA-related human casualties per unit of road length occur on the main roads with the highest AADT and the speed limit. In relation to WVA mitigation measures, 9.5% of fatalities and 1.4% of injuries were registered in areas with warning signs, and 10.4% of all casualties occurred in road segments protected by wildlife fencing. In a previous publication, we already pointed out that insufficient protection from WVAs might be related to the design and implementation of fencing [28]. It was concluded that “wildlife fencing … is shifting ungulate-wildlice accidents towards roads characterized by lower speed and traffic intensity… and re-allocates wildlife movement pathways toward roads with insufficient or no mitigation measures”.

Even though wildlife fences are considered one of the best protections against WVAs [21,33], their installation is subject to a number of requirements. These requirements aim not only to isolate animals from the road but also to guarantee their access to the other side of the road [16,23]. The location of the fences is typically chosen based on the number of WVAs and landscape features [14,27,70]. To the best of our knowledge, the installation of wildlife fences has never been planned according to the spatial pattern of human casualties.

It should be noted that even if all highways are fenced, WVAs still occur; e.g., in Hungary, 5% of traffic accidents involve wild animals [71]. The situation is similar in other European countries, where deer are the dominant wildlife species in WVAs [72,73,74]. However, data on human casualties in these countries were not discussed [71,72,73,74].

As summarized by Al-Bdairi et al. [32], although there is an extensive body of knowledge on animal–vehicle collisions, there is a very limited understanding of the relationship between driver injury severity resulting from such collisions and other factors. These factors include weather conditions, vehicle characteristics, roadway conditions, human-related factors such as driving style, and, finally, wildlife behavior [29,30,75,76,77].

In mid-latitude countries, trends in WVA are similar [65,66,76], even though drivers’ attitudes toward WVA may differ [30]. In the case of exotic animals, patterns might be different [78,79,80]. For example, drivers may swerve to avoid collisions with exotic species, such as the lowland tapir (*Tapirus terrestris*), southern tamandua (*Tamandua tetradactyla*), red-tailed boa (*Boa constrictor*) or blue-and-yellow macaw (*Ara ararauna*), more often than they do with common species.

Understanding the factors contributing to road traffic injuries is essential for enhancing road safety and implementing strategies to reduce both the frequency and severity of road-related accidents [81]. As demonstrated by Vanlaar et al. [76], drivers may either lack awareness of methods to avoid wildlife–vehicle collisions or, if aware, fail to comply with these methods. For instance, they might engage in abrupt braking or swerving instead of adopting safer measures such as slowing down and steering straight. Therefore, the role of the driver in WVA cannot be overestimated [30]. However, the topic has not been sufficiently covered thus far.

Wildlife–vehicle accidents occur at the intersection of roads and wildlife, resulting not only in human casualties and property loss but also in wildlife mortality and changes in their behavior [26]. Mitigation measures, especially wildlife fencing and passages, must prioritize the safety of multiple species [82] and address the risk to motorists’ safety [14]. In Lithuania, wildlife fencing should be designed primarily for moose [28].

The need for long-term data to identify priority sites emphasizes importance of public support and informed transportation policy [14,82,83]. However, roadkill mitigation programs are underfunded in Lithuania; therefore, informing drivers about the spatiotemporal distribution on WVAs, collision avoidance techniques and animal biology would be the most cost-effective way to improve road safety for wildlife.

Given that WVAs present various risks, including threats to human safety, property loss, and biodiversity, there are numerous advantages to be derived from minimizing their occurrences [84]. WVAs have the potential to impact one in every two drivers with one in four drivers facing property damage due to accidents involving animals [30]. These authors presume that by understanding and responding to drivers’ attitudes toward these incidents, the severity and frequency of WVAs can be reduced. Promoting safe avoidance methods could be encouraged through driver education, commencing with driving schools, and later on, incorporating training with simulators, as suggested by Vanlaar et al. [76]. Appropriate timing of driver education may improve road safety and reduce roadkills [84].

## 5. Conclusions

Our study aimed at analyzing human casualties in wildlife–vehicle accidents in Lithuania between 2002 and 2022. We conclude that human casualties constitute a minute proportion (<1% in 2018–2022) of WVAs, but still after 2015, every year, 21–48 persons suffer life-changing injuries. However, relying solely on exclusion measures proves insufficient, as approximately 10% of casualties occur within fenced road segments.

Given that moose are responsible for 66.7% of fatalities and 47.2% of injuries, mitigation measures addressing WVAs, including population abundance regulation, should predominantly focus on this species.

A potentially effective mitigation strategy may include enhanced information dissemination for drivers, taking into account the temporal and spatial patterns of casualties in WVAs.

## Figures and Tables

**Figure 1 animals-14-01452-f001:**
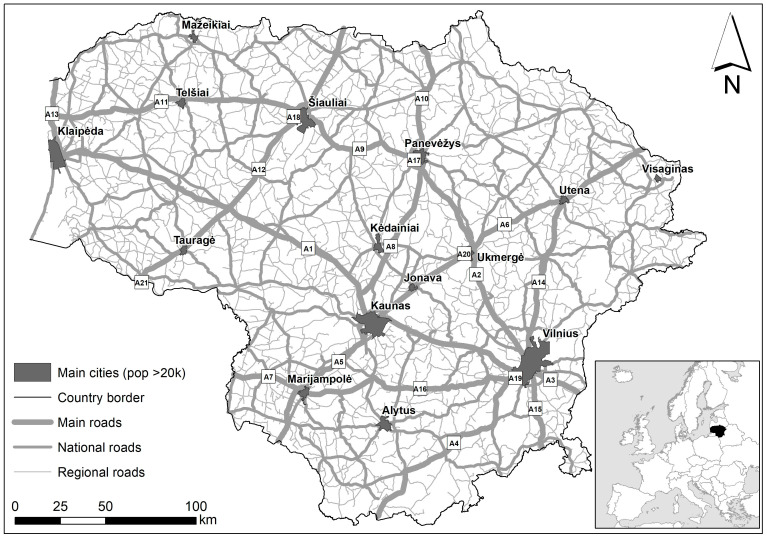
Study area with road type and main cities indicated.

**Figure 2 animals-14-01452-f002:**
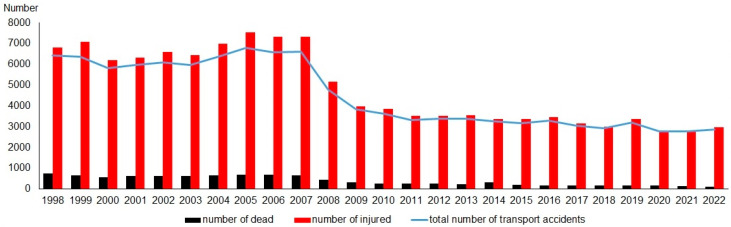
The number of transport accidents in Lithuania during the 1998–2022 period, as registered by the Lithuanian Road Police Service and related number of human casualties, according to [46].

**Figure 3 animals-14-01452-f003:**
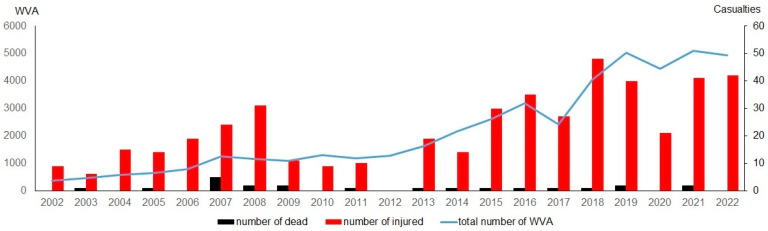
Number of wildlife–vehicle collisions in Lithuania during the 1998–2022 period as registered by the Lithuanian Police Traffic Surveillance Service, and related number of human casualties.

**Figure 4 animals-14-01452-f004:**
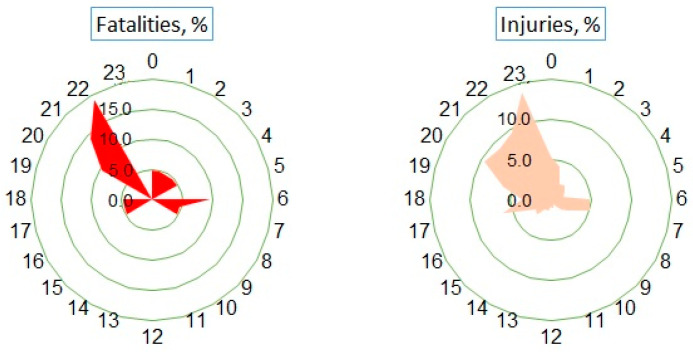
Distribution of human casualties by hour of the day.

**Figure 5 animals-14-01452-f005:**
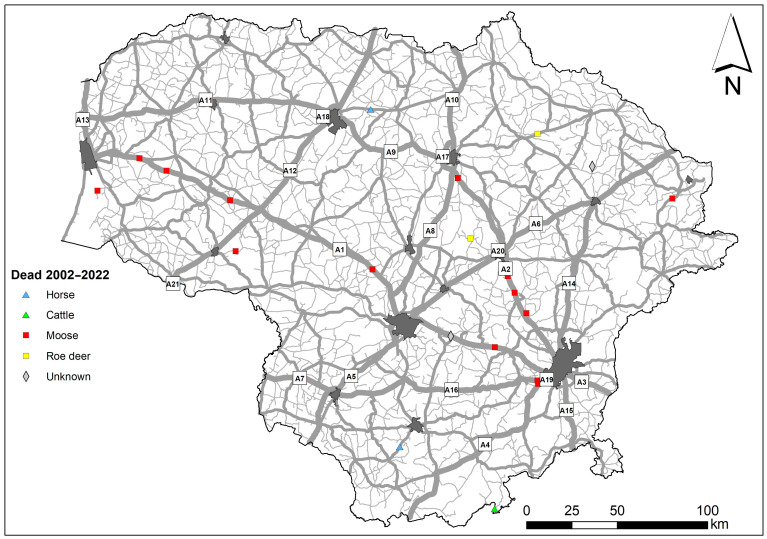
Spatial distribution of human fatalities in WVAs on the roads of Lithuania. Main road numbers represented in square labels.

**Figure 6 animals-14-01452-f006:**
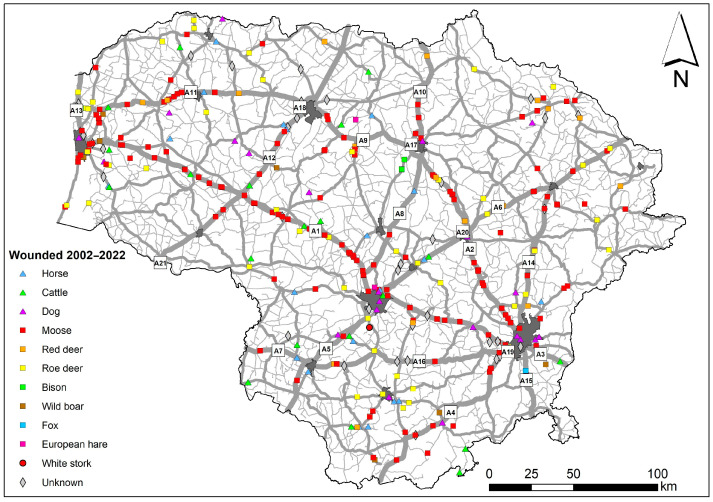
Spatial distribution of human injuries in WVAs on the roads of Lithuania. Main road numbers represented in square labels.

**Figure 7 animals-14-01452-f007:**
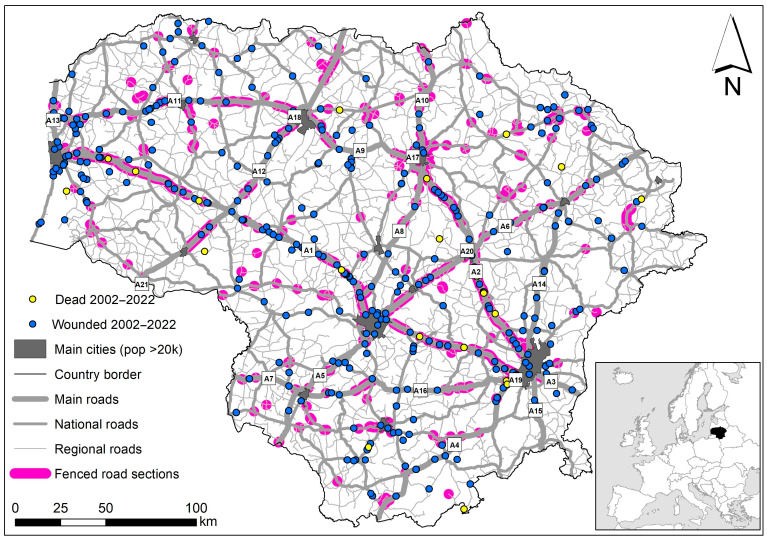
The occurrence of human casualties in WVAs with relation to the presence of wildlife fencing. Fenced segments represent situations after 2017; therefore, some casualties occurred before the construction of fencing.

**Table 1 animals-14-01452-t001:** Wildlife and domestic animal species involved in WVA resulting in human casualties in Lithuania from 2002 to 2022.

Species	Body Mass, kg	Fatalities		Injuries		
		WVA Number	N	WVA Number	N	Per 1 WVA
European bison (*Bison bonasus*)	700			4	7	1.75
Moose (*Alces alces*)	325	14	14	170	217	1.28
Red deer (*Cervus elaphus*)	105			17	21	1.24
Roe deer (*Capreolus capreolus*)	28	2	2	49	51	1.04
Wild boar (*Sus scrofa*)	120			9	10	1.11
Red fox (*Vulpes vulpes*)	5.3			1	1	1.00
European hare (*Lepus europaeus*)	4.6			2	2	1.00
White stork (*Ciconia ciconia*)	4.5			1	1	1.00
Horse (*Equus ferus caballus*)	600	2	3	22	32	1.45
Cattle (*Bos taurus*)	400	1	1	23	34	1.48
Cat (*Felis catus*)	4			1	1	1.00
Dog (*Canis familiaris*)	10			21	23	1.10
Species not identified		2	2	40	52	1.30

**Table 2 animals-14-01452-t002:** Distribution of human casualties by month.

Casualty	Jan	Feb	Mar	Apr	May	Jun	Jul	Aug	Sep	Ost	Nov	Dec
Fatality	1		1	1	3	3		4	5		2	1
Injury	14	8	9	23	55	44	39	27	53	42	26	16
Total	15	8	10	24	58	47	39	31	58	42	28	17
%	4.0	2.1	2.7	6.4	15.4	12.5	10.3	8.2	15.4	11.1	7.4	4.5

**Table 3 animals-14-01452-t003:** Distribution of human casualties by day of the week.

Casualty	Monday	Tuesday	Wednesday	Thursday	Friday	Saturday	Sunday
Death	1	4	4	1	3	1	7
Injury	35	47	50	43	62	59	60
Total	36	51	54	44	65	60	67
%	9.6	13.6	14.4	11.8	17.4	16.0	17.9

**Table 4 animals-14-01452-t004:** Distribution of human casualties by road type. Superscript letters indicate differences between the values in the columns with significance at *p* < 0.05.

Road	Casualty			Per 1000 km
	Fatality	Injury	Total	
Main	11 ^a^	154 ^a^	165	94.25
National	4 ^a^	112 ^b^	116	23.54
Regional	6 ^a^	32 ^c^	38	2.61
Other	0	58 ^d^	58	n/a

**Table 5 animals-14-01452-t005:** Human injuries in WVAs on the main roads of Lithuania and road characteristics.

Main Road		AADT in 2016			WVA with Human Injuries		
Number	Length, km	Cars	Trucks	Total	n	%	per km
A1	311.4	18,937	2895	21,832	51	33.1	0.164
A2	135.9	10,910	1230	12,140	21	13.6	0.155
A11	146.85	5166	580	5746	17	11.0	0.116
A6	185.4	6181	876	7057	12	7.8	0.065
A12	186.09	4669	547	5216	9	5.8	0.048
A4	134.46	4720	389	5109	8	5.2	0.059
A14	95.6	6584	435	7019	7	4.5	0.073
A16	137.51	5307	589	5896	6	3.9	0.044
A13	45.15	10,842	701	11,543	5	3.2	0.111
A10	66.1	7776	2408	10,184	4	2.6	0.061
A8	87.86	7038	2504	9542	3	1.9	0.034
A9	78.94	7667	896	8563	3	1.9	0.038
A5	97.06	15,078	5280	20,358	2	1.3	0.021
A7	42.21	4595	560	5155	2	1.3	0.047
A15	49.28	5428	548	5976	2	1.3	0.041
A17	22.28	8029	2708	10,737	1	0.6	0.045
A3	33.99	5863	917	6780	1	0.6	0.029

**Table 6 animals-14-01452-t006:** Driving-related determinants of human casualties during wildlife–vehicle accidents: 1—Hit an animal; 2—Ran off the road; 3—Hit a tree; 4—Hit another car; 5—Failed to brake; 6—Unsuccessful swerve in attempt to avoid collision; 7—Swerved into opposite lane; 8—Hit by another car; 9—Hit a guardrail; 10—Car rolled over; 11—Hit a pole; 12—Drunk driving.

Fatal Casualties		Injuries	
Sequence	N	Sequence	N
1	13	1, 6	1
1, 2	1	1, 6, 10	1
1, 2, 10	2	1, 7, 4	2
1, 3	2	1, 8, 4	1
1, 7, 4	1	1, 9	2
1, 7, 10	1	1, 10	7
1, 9, 8	1	1, 12	2
Injuries		2, 1	1
1	280	5	4
1, 2	14	5, 2	1
1, 2, 3	9	5, 2, 3	1
1, 2, 10	4	5, 2, 10	1
1, 2, 9	3	5, 10	3
1, 2, 11	1	6, 2	1
1, 3	2	6, 2, 3	1
1, 4	1	6, 2, 10	1
1, 5	1	6, 7, 4	1
1, 5, 10	1	Other	9

## Data Availability

Due to the sensitive nature of the study, raw data would remain confidential and would not be shared.

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
