# Peer review of "Trends and Characteristics of Human Casualties in Wildlife–Vehicle Accidents in Lithuania, 2002–2022"

_animals, 2024, doi:10.3390/ani14101452_

Round 1
Reviewer 1 Report
Comments and Suggestions for Authors
An interesting overview of wildlife-vehicle accidents in a small but highly forested European country showing that cervid populations are an increasing hazard in road and driver safety.
Line 38: replace 'toll is' with cause
Line 39: replace 'got' with sustain and rephrase to read 'reduce human mortality on roads'
Line 40: insert 'the' before 'number'
Line 42: plural 'accidents'
Line 43: insert 'few' after 'last'
Line 55: insert 'The' at start of sentence
Line 65: reword 'the number of human casualties unreported'
Line 66: insert 'in' after resulted
Line 67: define 'Baltic countries' - do you mean Lithuania, Latvia and Estonia or all countries surrounding the Baltic Sea? Replace 'in' with 'is'
Line 85: replace 'in' with 'a'
Line 91: delete first 'the'
Line 104: insert 'sizes' after sample
Line 105: suggest 'do not provide a WVA category, although....'
Line 109: suggest 'mention WVA problems'
Line 114: The aim of this paper....
Line 121: suggest 'Our study......, a small country in Northern Europe'
Line 122: replace 'the' with 'a' x3
Line 124 onwards: for consistency I suggest you stick to a single decimal place for your percentages
Lines 127-129: replace 'long' with 'of' x3
Line 131: suggest 'mostly installed by 2017'
Line 134: replace 'implemented' with 'constructed'
Line 135: 'one-way' gates rather than one-sided? What is a 'deer crate' and explain how they work
Line 141: suggest 'Recording the number of .......was started by Lithuanian police in .....'
Line 145: suggest 'in the on site reports, but no check of of the fate of injured persons in hospitals was made.'
Line 153: explain what the purpose of 'wildlife enclosures' is. Are they fenced reserves?
Line 154: delete 'For the confidence interval' and start new sentence with 'We'
Line 163: start with 'The'
Line 164: replace 'in the date' with 'at the time'
Line 167: replace 'illuminance' with 'light intensity'
Line 173: suggest use of 'geo-referenced' rather than 'geo-localized'
Line 195: it is not a spike but a drop (or downward spike)
Line 212: replace 'culprit' with 'cause'
Table 1, first column: left-hand justify the species
Line 299: suggest 'occurred prior to the fence construction' and perhaps use construction rather than implemented in the following paragraphs
Line 367: again, replace 'culprit' with 'cause'
Line 368-369: are the numbers the national population estimates for the two species? Are populations stable, increasing or declining?
Line 379: in this paragraph you could discuss whether the May and September 'spikes' correspond to significant events in the species annual cycle or movement ecology
Line 391: if most accidents occur on main roads, does this correspond to vehicle speed? Are there different speed limits on main, national and regional roads?
Line 396: perhaps briefly summarise the main findings of reference 28
Line 410: define 'exotic' animals, give some examples
Line 413: replace 'Comprehending' with 'Understanding'
Line 425: again, are wildlife enclosures the same as fenced reserves
Line 439: suggest reword ' Appropriate timing of driver education may improve road safety and reduce roadkills'.
Line 444: maybe replace 'health impairments' with 'life-changing injuries'?
Comments on the Quality of English Language
Please read your manuscript carefully and insert definite ('the') and indefinite ('a', 'an') articles, particularly in the Introduction and Methods sections. I refer to a some instances where this is needed in my specific comments above but there are others.
Overall, the quality of English is satisfactory and the article is readable.
Author Response
Reviewer #1 comments and answers
An interesting overview of wildlife-vehicle accidents in a small but highly forested European country showing that cervid populations are an increasing hazard in road and driver safety.
Line 38: replace 'toll is' with cause
- done
Line 39: replace 'got' with sustain and rephrase to read 'reduce human mortality on roads'
- done
Line 40: insert 'the' before 'number'
- done
Line 42: plural 'accidents'
- changed
Line 43: insert 'few' after 'last'
- inserted
Line 55: insert 'The' at start of sentence
- inserted
Line 65: reword 'the number of human casualties unreported'
- reworded
Line 66: insert 'in' after resulted
- inserted
Line 67: define 'Baltic countries' - do you mean Lithuania, Latvia and Estonia or all countries surrounding the Baltic Sea? Replace 'in' with 'is'
- inserted names of the three countries
Line 85: replace 'in' with 'a'
- replaced
Line 91: delete first 'the'
- deleted
Line 104: insert 'sizes' after sample
- inserted
Line 105: suggest 'do not provide a WVA category, although....'
- changed as suggested
Line 109: suggest 'mention WVA problems'
- changed as suggested
Line 114: The aim of this paper....
- inserted
Line 121: suggest 'Our study......, a small country in Northern Europe'
- changed as suggested
Line 122: replace 'the' with 'a' x3
- changed as suggested
Line 124 onwards: for consistency I suggest you stick to a single decimal place for your percentages
- decimal places truncated
Lines 127-129: replace 'long' with 'of' x3
- changed as suggested
Line 131: suggest 'mostly installed by 2017'
- changed as suggested
Line 134: replace 'implemented' with 'constructed'
- changed as suggested
Line 135: 'one-way' gates rather than one-sided? What is a 'deer crate' and explain how they work
- yes, one-way. Deer crates – sorry, failed to find proper English word. It is “wildlife guard” - Cattle guards, modified to discourage wild ungulates from crossing, or wildlife guards can be embedded in the road surface at fence ends. They typically consist of a pit with metal bars or bridge grate material on top.
Line 141: suggest 'Recording the number of .......was started by Lithuanian police in .....'
- changed as suggested
Line 145: suggest 'in the on site reports, but no check of of the fate of injured persons in hospitals was made.'
- changed as suggested
Line 153: explain what the purpose of 'wildlife enclosures' is. Are they fenced reserves?
- again sorry, we meant “wildlife fencing”
Line 154: delete 'For the confidence interval' and start new sentence with 'We'
- changed as suggested
Line 163: start with 'The'
- inserted
Line 164: replace 'in the date' with 'at the time'
- changed as suggested
Line 167: replace 'illuminance' with 'light intensity'
- replaced as suggested
Line 173: suggest use of 'geo-referenced' rather than 'geo-localized'
- changed as suggested
Line 195: it is not a spike but a drop (or downward spike)
- changed as suggested
Line 212: replace 'culprit' with 'cause'
- changed as suggested
Table 1, first column: left-hand justify the species
Rebuttal: this is MDPI style, would be done at layout phase if required
Line 299: suggest 'occurred prior to the fence construction' and perhaps use construction rather than implemented in the following paragraphs
- changed as suggested
Line 367: again, replace 'culprit' with 'cause'
- changed as suggested
Line 368-369: are the numbers the national population estimates for the two species? Are populations stable, increasing or declining?
- yes, these are national estimates, and both are increasing
Line 379: in this paragraph you could discuss whether the May and September 'spikes' correspond to significant events in the species annual cycle or movement ecology
- explanation added
Line 391: if most accidents occur on main roads, does this correspond to vehicle speed? Are there different speed limits on main, national and regional roads?
- yes, speed limit is highest on the main rods, we added this information
Line 396: perhaps briefly summarise the main findings of reference 28
- brief conclusions added
Line 410: define 'exotic' animals, give some examples
- examples included
Line 413: replace 'Comprehending' with 'Understanding'
- replaced
Line 425: again, are wildlife enclosures the same as fenced reserves
- changed to “fencing”
Line 439: suggest reword ' Appropriate timing of driver education may improve road safety and reduce roadkills'.
- changed as suggested
Line 444: maybe replace 'health impairments' with 'life-changing injuries'?
- changed as suggested
The authors sincerely appreciate your help and corrections. All of them have been accepted except for the alignment of the table. The latter is part of the MDPI style, and we prefer to allign with it.
Reviewer 2 Report
Comments and Suggestions for Authors
Dear Authors,
I have found your manuscript well prepared and providing valuable information.
1. I suggested some minor changes in the text (pdf) and a better presentation for some numbers that were difficult to imagine.
2. At the end of the references, I added a few relevant European papers that can be cited in your text.
Best regards,

The English is good, but a native speaker can certainly improve it!
Author Response
Reviewer #2 comments and answers
Dear Authors,
I have found your manuscript well prepared and providing valuable information.
Comment 1. I suggested some minor changes in the text (pdf) and a better presentation for some numbers that were difficult to imagine.
Answer: thank you for your work. We incorporated requested changes, however, there were some disputable moments we need to mention.
- Line 195: you say growth of WVA number is logistic, not linear, and then show this on the Figure 3. Sorry, curve you draw approximated numbers of injuries. Number of WVA is not saturated – in 2023 there were over 6000 WVA registered, therefore growth is really linear;
- Line 205: recalculated per 1000 WVA, and changed in the text;
- Line 291: comment “Put table header rows on the top of the continuing table” is not relevant, as MDPI is not using this feature in their layout. Position of the Table will be different, and no changes will be required;
- Line 352: as we say, increase of WVA number is really linear, not logistic. As we understand, in the case of logistic regression, explanation of the saturation level should be given, but there is no saturation of WVA number’
Comment 2. At the end of the references, I added a few relevant European papers that can be cited in your text.
Answer: last of your recommended to cite papers is duplicating information from the fourth one, it deals with the same M3 highway in Hungary, therefore we skip it. The rest we found possible to cite, though none of them is directly related to the topic of out paper – human injuries and fatalities. Therefore, these four papers are citable as mentioning roe deer – vehicle collisions in different EU countries. We used these sources in Discussion.
Reviewer 3 Report
Comments and Suggestions for Authors
This manuscript focuses on human trauma as result of wildlife vehicle accidents (WVA) in Lithuania. This is unusual for a biological journal and so it is beholden on the authors to tease out causes in relation to the wildlife. The authors do a fair job of this identifying moose as the primary cause of human mortality in accidents and noting that this is disproportionate to both their abundance and the frequency of WVA with ungulates. They further examine whether measures such as fencing are mitigating and find some failure in this regard. They examine the sequence of events leading to a human fatality in a WVA and make some inferences about driver behaviour, but little direct evidence is provided and as such is a limitation of this study, especially as a recommendation from the study is driver education to respond to or avoid impending WVA. The authors refer to a potential discontinuity in their data set given it spans responses to the COVID-19 pandemic. It would be useful to include specific measures employed in Lithuania (and their duration) and how they impacted traffic volumes.
The manuscript amalgamates two writing styles, one with correct use of definite/indefinite articles (i.e. ‘the’ and ’a/an’) in English grammar. The other ignores their use. The introduction and materials and methods are particularly problematic. Both these sections need extensive editing of the grammar and expression. It would be very tedious to list all the errors. The rest of the text is well-written.
In summary, the study is interesting but not ground-breaking. The analysis of the results is simple but appropriate given the small sample size of human mortality in WVA. More could be made of the Lithuanian context such as a relatively small road network accessed by presumably sophisticated users in the heartland of the EU. This might focus the authors’ recommendations for further research.
Comments on the Quality of English LanguageExtensive editing of the 'Introduction' and 'Materials and Methods' is required. Most of the errors relate to lack of use of definite/indefinite articles but there are some other grammatical errors and poor expression.
Author Response
eviewer #3 comments and answers
This manuscript focuses on human trauma as result of wildlife vehicle accidents (WVA) in Lithuania. This is unusual for a biological journal and so it is beholden on the authors to tease out causes in relation to the wildlife. The authors do a fair job of this identifying moose as the primary cause of human mortality in accidents and noting that this is disproportionate to both their abundance and the frequency of WVA with ungulates. They further examine whether measures such as fencing are mitigating and find some failure in this regard. They examine the sequence of events leading to a human fatality in a WVA and make some inferences about driver behaviour, but little direct evidence is provided and as such is a limitation of this study, especially as a recommendation from the study is driver education to respond to or avoid impending WVA. The authors refer to a potential discontinuity in their data set given it spans responses to the COVID-19 pandemic. It would be useful to include specific measures employed in Lithuania (and their duration) and how they impacted traffic volumes.
Answer: COVID-19 pandemic and WVA, as well as limitation of human mobility were already discussed in two our publications, so we found impossible to discuss this topic repeatedly. On the other hand, additional numeric information of limiting measures is not available.
Comment: The manuscript amalgamates two writing styles, one with correct use of definite/indefinite articles (i.e. ‘the’ and ’a/an’) in English grammar. The other ignores their use. The introduction and materials and methods are particularly problematic. Both these sections need extensive editing of the grammar and expression. It would be very tedious to list all the errors. The rest of the text is well-written.
Answer: please see changes in the two chapters you mention, and thank you for the pointing this out. Reviewer 1 also noticed problems in the Introduction and Materials and methods. Hope we improved this text.
Comment: In summary, the study is interesting but not ground-breaking. The analysis of the results is simple but appropriate given the small sample size of human mortality in WVA. More could be made of the Lithuanian context such as a relatively small road network accessed by presumably sophisticated users in the heartland of the EU. This might focus the authors’ recommendations for further research.
Answer: unfortunately, road management in Lithuania currently is not the best one. Therefore, before recommending changes, we need to do lobbying with responsible institutions. Scientific publications is a good basis for the public involvement by popular publications – thus, this paper will be used as a basis to work forther.
Comments on the Quality of English Language: Extensive editing of the 'Introduction' and 'Materials and Methods' is required. Most of the errors relate to lack of use of definite/indefinite articles but there are some other grammatical errors and poor expression.
Answer: please see changes in the two chapters you mention.